# Indoor and outdoor fine particulate matter and carbon monoxide concentrations in homes of infants in Nairobi, Kenya

Vincent K. Kipter[1], Faridah H. Were[1], Michael J. Gatari[2], Christopher Zuidema[3], Edmund Y. W. Seto[3], Orly D. Stampfer[3‡], Barbra A. Richardson[4,5], Bernard Makau[6‡], Priscilla Wanini Edemba[6‡], Emily Adhiambo[6‡], Julian D. Marshall[7], Timothy V. Larson[3,7], Catherine J. Karr[3,8‡], Elizabeth Maleche-Obimbo[9‡], Sarah Benki-Nugent[4‡], Anne M. Riederer[3‡*]

**1** Department of Chemistry, Faculty of Science & Technology, University of Nairobi, Nairobi, Kenya, **2** Institute of Nuclear Science & Technology, University of Nairobi, Nairobi, Kenya, **3** Department of Environmental & Occupational Health Sciences, University of Washington, Seattle, Washington, United States of America, **4** Department of Global Health, University of Washington, Seattle, Washington, United States of America, **5** Department of Biostatistics, University of Washington, Seattle, Washington, United States of America, **6** Kenyatta National Hospital, Nairobi, Kenya, **7** Department of Civil & Environmental Engineering, University of Washington, Seattle, Washington, United States of America, **8** Department of Pediatrics, University of Washington, Seattle, Washington, United States of America, **9** Department of Paediatrics and Child Health, University of Nairobi, Nairobi, Kenya

☯ These authors contributed equally to this work.
‡ These authors also contributed equally to this work.
* anneried@uw.edu

## Abstract

Early life exposure to air pollution is associated with adverse health outcomes in children however few studies have investigated children's air pollution exposures in urban settings in sub-Saharan Africa (SSA). We measured fine particulate matter (PM$_{2.5}$) and carbon monoxide (CO) in homes of infants in Nairobi, Kenya and conducted exploratory analysis of exposure factors. Questionnaires captured household characteristics and self-reported air pollution exposures. Indoor and outdoor 24-hour (24 h) concentrations were measured inside and 1 m outside the house. PM$_{2.5}$ was sampled using standard gravimetric procedures; CO was measured with direct-reading electrochemical sensors. Forty-eight homes were sampled at median infant age 11.5 months (range 0.8-26.2 months). During sampling, 66.7%, 18.8%, 10.4% and 10.4% of mothers, respectively, reported using liquefied petroleum gas (LPG), ethanol, electricity, and kerosene for cooking. Median indoor and outdoor 24 h PM$_{2.5}$ concentrations (n = 39) were 39.9 ug/m³ (range, 12.8-519.6 ug/m³) and 23.3 ug/m³ (range, 2.6-68.2 ug/m³), respectively. Most PM$_{2.5}$ concentrations (97% of indoor; 79% of outdoor) exceeded the World Health Organization (WHO) 24 h air quality guideline (AQG) of 15 ug/m³. Median indoor (n = 47) and outdoor (n = 41) 24 h mean CO concentrations were 0.7 ppm (range, 0-33.9 ppm) and 0.0 ppm (range, 0-1.0 ppm), respectively. Mean indoor CO concentrations exceeded the WHO 24 h AQG of 6.2

**Data availability statement:** The deidentified data that support the findings of this study are available in the supporting information.

**Funding:** Funding for the Kenya Healthy Home Healthy Brain Pilot Project and the Air Pollution and Brain Development (ABC) Study was provided by the U.S. National Institutes of Environmental Health Sciences (R01ES032153 to SBN and P30ES007033 to Terrence Kavanagh). The funders had no role in study design, data collection and analysis, decision to publish, or preparation of the manuscript.

**Competing interests:** The authors have declared that no competing interests exist.

ppm in 9% of homes. Despite frequent use of cooking fuels considered to be clean such as LPG and ethanol, $PM_{2.5}$ and CO levels in infant homes in urban SSA often exceeded the WHO AQGs. Expanded studies of children's air pollution exposures in urban SSA are needed to build awareness and inform policy.

## Introduction

Early life exposure to combustion-related air pollutants such as particulate matter ≤ 2.5 um aerodynamic diameter ($PM_{2.5}$) is associated with adverse health outcomes in children [1,2]. Elevated levels of $PM_{2.5}$, carbon monoxide (CO), and other household air pollutants, from burning poorly ventilated, high-emitting cooking and lighting fuels such as biomass and kerosene, have been documented in both rural and urban sub-Saharan Africa (SSA) [2–8]. Although some urban households in Kenya and other SSA countries may have access to clean fuels, such as electricity, ethanol, and liquefied petroleum gas (LPG) [2,9], urban children are exposed to dirty household cooking fuels in addition to outdoor (ambient) air pollution [2]. The sources of ambient air pollution in urban SSA settings include poorly regulated industrial operations in residential areas, rubbish burning, and unpaved and highly trafficked roadways [2,8,10]. In naturally ventilated homes (i.e., homes without fans or other mechanical ventilation), such as those common in SSA, outdoor air pollution can affect indoor air quality. However, indoor air pollutants can also accumulate, even in homes using clean fuels, when cooking appliances are not properly vented (i.e., exhausted to the outdoors) and/or windows and doors are kept closed during cooking or other combustion activities.

Few studies to date have measured $PM_{2.5}$ and CO in non-biomass burning children's homes in urban SSA. Shupler et al. (2024) measured 24 h $PM_{2.5}$ and CO concentrations in kitchen air and personal air of mothers (n = 248) and children (n = 124) in 256 homes in peri-urban communities (i.e., urbanizing areas adjacent to city centers) in Kenya, Cameroon and Ghana, and outdoor 24 h $PM_{2.5}$ at several sites in each community [2]. Geometric mean concentrations of the 24 h $PM_{2.5}$ and CO in kitchen and personal air from biomass burning households were indeed higher than those using LPG [2]. However, even LPG using households had kitchen $PM_{2.5}$ concentrations exceeding the World Health Organization (WHO) 24 h air quality guideline (AQG) (15 ug/m³), an outdoor limit designed to protect the general population from excess mortality from air pollution [11]. This may reflect infiltration of ambient air pollution indoors, particularly in the Ghana community where ambient $PM_{2.5}$ was higher (mean 31 ug/m³) compared to that in Cameroon (mean 14 ug/m³) and Kenya (mean 6 ug/m³) communities [2,11]. Shezi et al. (2018) found a mean 24 h $PM_{2.5}$ of 38 ug/m³ (median 28 ug/m³) in 300 homes of low-income mothers and children in Durban, South Africa even though most (93%) reported using electricity for cooking [5]. In bivariate analyses, they found higher $PM_{2.5}$ in homes with indoor combustion activities (cooking, smoking, burning incense/candles) during sampling compared to none, and higher $PM_{2.5}$ in homes ≤ 1.6 km from a major road compared to those > 1.6

km away [5]. This illustrates the importance of both indoor and outdoor sources on indoor $PM_{2.5}$ in urban SSA children's homes.

To our knowledge, no studies to date have measured both indoor and outdoor $PM_{2.5}$ and CO in the homes of young children in urban SSA [2,8,10,12,13]. Measurements of indoor and outdoor pollutant concentrations allow for the quantification of the indoor/outdoor (I/O) ratio, an important parameter for both indoor air quality and epidemiologic studies. The I/O ratio, as its name suggests, is the ratio of measured indoor to outdoor air pollution concentrations, and is commonly used to understand the relative importance of indoor vs. outdoor environments on human exposures to particulate air pollution, which are due in part to indoor concentrations vs. outdoor concentrations as well as time spent indoors vs. outdoors [14,15]. Studies have measured $PM_{2.5}$ I/O ratios in a variety of residential settings [16–27]. The ratio has been found to be less than one in some studies where homes are well sealed, have air filtration or few indoor sources; but it has also been found to be greater than one in some studies where indoor smoking and combustion sources are present [14,24,25,26,27]. A previous study reporting both indoor and outdoor $PM_{2.5}$ concentrations exists for the N'gando informal settlement community in Nairobi, Kenya [4]. A few studies have measured indoor air quality in Kenya [28]; but there is a lack of $PM_{2.5}$ I/O data on urban communities in Nairobi, and data collected in the context of pregnancy and infancy are lacking. As part of a novel longitudinal pregnancy cohort study to understand exposure to air pollution and neurodevelopmental health among children in Nairobi, Kenya [11,29,30], we measured 24 h indoor and outdoor $PM_{2.5}$ and CO in a subset of infant homes. We also explored associations between the measured concentrations and household combustion activities and ventilation characteristics, both key influences on indoor air quality [31]. This study contributes empirical evidence on air pollution exposures among urban SSA infants. Such data are needed to help design interventions to reduce exposures and address inequities in morbidity [11,13,30].

## Methods

### Study population

The Air Pollution Exposures in Early Life and Brain Development in Children (ABC) Study is a prospective pregnancy cohort study which aims to test the association between prenatal exposure to air pollution and neurodevelopmental outcomes in Nairobi, Kenya [32]. Briefly, a convenience sample of N = 400 pregnant women attending routine antenatal care visits was enrolled from January 10 to November 29, 2022 from the Dandora II Health Centre, a publicly accessible health facility near the Dandora dump, a > 30-acre dumpsite with routine rubbish burning. Key eligibility criteria were pregnant and maternal age 18–40 years. To enable exploratory modeling of early life air pollution exposures, a convenience subset of 48 ABC participants was invited to participate in an air sampling home visit (described below). During the recruitment phase of the study, one to two women were invited for this air sampling visit each week. There were no pre-specified exclusion criteria for participation in air sampling.

Written informed consent for study procedures was obtained from study participants. Study procedures were approved by the Kenyatta National Hospital/University of Nairobi Ethics and Research Committee and the University of Washington Human Subjects Division.

### Enrollment and home visit interviews

At enrollment, a study clinician interviewed participants using a structured questionnaire. The questionnaire included questions on maternal demographics (e.g., age, education, employment, household characteristics related to indoor air pollution exposures and ventilation (e.g., number of rooms, people per room, and number of doors, windows and vents), and household and other daily behaviors involving combustion (e.g., use of cooking fuel, candles, incense, mosquito coils or repellent, rubbish burning, cigarette and marijuana smoking) and their frequency (daily, most days, some days, rarely, not at all). Fuels included electricity, LPG, paraffin (kerosene), ethanol-based fuel, wood, and charcoal.

During the air sampling set-up and take-down visits, participants were interviewed and reported the following: sources of indoor and outdoor air pollution (e.g., fuel use indoors and outdoors; own or neighbors' outdoor cooking smoke (< 20 m); and sources near (< 1 km) the home including vehicles exhausts, dumpsite, factories, rubbish burning, charcoal or brick making, dusty roads, construction dust, and burning tires. At the take-down visit, participants were interviewed on indoor combustion activities (e.g., use of cooking fuels, incense, candles, kerosene lamp(s), mosquito repellent, and rubbish burning), and outdoor air pollution observed near the home during the prior 24 hours. A laser measure (GLM 20, Bosch GmbH, Gerlingen-Schillerhöhe, Germany) was used to measure kitchen dimensions and distance from the outdoor sampling equipment to the home's closest outer wall.

## PM$_{2.5}$ measurements

Twenty-four-hour PM$_{2.5}$ indoor and outdoor samples were collected at participating homes using Harvard Impactors loaded with 37 mm, 2 um pore size, polytetrafluoroethylene filters (SKC, Eighty Four, Pennsylvania, USA), drawn by SKC AirChek-XR5000 pumps with a flow rate of 1.8 L/min. Indoor samples were collected in the kitchen/cooking area and outdoor samples were collected approximately 1 m from the outer wall of the home, at breathing height (i.e., ~ 1.8 m from the ground) in both cases. Pumps were calibrated by the manufacturer to U.S. National Institute of Standards and Technology (NIST) traceable standards and flow rates were checked at deployment and take-down using a DryCal® DC-Lite Medium flow meter (Mesa Laboratories, Lakewood, CO, USA) also calibrated to NIST traceable standards. Samples with flow rates of 1.4-2.2 L/min were considered acceptable. Impactors were cleaned, wrapped in aluminum foil, and stored in zipper lock bags between deployments and nitrile gloves were worn during all sampling and filter handling activities.

Filters were pre- and post-conditioned and weighed following U.S. Environmental Protection Agency (EPA) protocols [33] in a temperature and humidity-controlled laboratory at the Department of Chemistry, University of Nairobi, using a Shimadzu AUW220D semi-microbalance (Shimadzu Corporation, Kyoto, Japan). The balance sensitivity is 10 ug minimum display and a repeatability (SD [standard deviation]) of ≤ 50 ug. However, we assessed repeatability of the balance as ± 20 ug using triplicate weights of 50 mg and 100 mg NIST traceable working mass standards. Based on this, ± 20 ug (for each of the triplicate weights) was used as the acceptance criterion for valid filter mass. Temperature (T) and relative humidity (RH) in the laboratory were monitored continuously using a Lascar EasyLog EL-WiFi-TH Temperature & Humidity Data Logger (Lascar Electronics, Erie, PA, USA). In general, the difference in RH between pre- and post-weighing was ≤ 3% for each sample, with a range of 0–20%. RH differences between pre- and post- filter weighing were considered in terms of their possible impact on concentration uncertainties. Percentage differences ≥ 10% were considered to have a non-negligible impact on the uncertainties and the related filters were therefore rejected.

Filters were inspected for holes or tears before deployment and again before weighing, and defective filters were rejected. The assembled impactors with sample filters were wrapped with aluminum foil and transported to the sampling site in a tightly sealed cooler box. Three blank filters (two from the laboratory and a field blank) were analyzed for each batch of 10 filters. A field blank was a primary sample filter loaded on the impactor, carried along with other filters and not used for sampling. Control charts of field and laboratory blank mass differences (post minus pre weight) revealed no apparent pattern. The method detection limit, calculated as three times the median absolute deviation of the field blanks mass differences (n = 30), was 21 ug. Two outdoor samples were below this value; the machine values were used for these samples in data analyses. Filter masses were calculated as the mean of triplicate weights and filter mass difference was calculated as the post minus the pre filter mass. Mass differences were divided by the volume of air sampled to calculate the PM$_{2.5}$ concentrations. Two pairs of collocated indoor and one pair of collocated outdoor duplicate samples were collected, and precision was calculated as the relative percent difference between duplicates concentrations. Finally, to evaluate how much the indoor home space is protected from infiltration of outdoor air pollutants, PM$_{2.5}$ I/O ratios were calculated by dividing the indoor by the outdoor concentration for homes with complete measurements. A lower ratio indicates more protection [16,19–22,34,35].

## CO measurements

Indoor and outdoor CO was measured using electrochemical EL-USB-CO sensors (Lascar Electronics Ltd., Whiteparish, UK) logging at 10 sec intervals and collocated with the $PM_{2.5}$ impactors. The manufacturer-rated measurement range is 0.0-300 ppm with a 0.5 ppm resolution [36]. Sensor data were checked for error messages and smoothed 1 min medians calculated. Precision was assessed by examining the slope, intercept, $R^2$, and root mean square error of bivariate regressions of 1 min CO data over the 24 h sample period between two collocated indoor and one collocated outdoor duplicate pair. The manufacturer stated accuracy is an overall error of ± 5 ppm or ± 4%.

Indoor and outdoor T and RH were also measured using HOBO® U12 Temp/RH Data Loggers (Onset, Bourne, MA, USA). These were collocated with the $PM_{2.5}$ and CO monitors and logged at 5 min intervals.

## Statistical methods

Statistical analyses were conducted in R version 4.3.3 (R Foundation for Statistical Computing, Vienna, Austria). For the sensor data, descriptive statistics were calculated for the 24 h sampling period using 1 min smoothed (CO) or 5 min logged measurements (T and RH) for each participant. Time series of indoor and outdoor CO were plotted for each home using the 1 min smoothed data.

We assessed the normality of the 24 h measurements by visually inspecting histograms of raw and natural log transformed concentrations. We calculated descriptive statistics, Spearman correlation coefficients ($\rho$) between the indoor and outdoor $PM_{2.5}$ measurements and between the indoor $PM_{2.5}$ and CO measurements, and $PM_{2.5}$ I/O ratios. We also compared indoor and outdoor $PM_{2.5}$ and CO concentrations to the WHO 24 h AQGs. The 24 h WHO AQG for $PM_{2.5}$ (15 ug/m$^3$) is primarily an ambient limit (although it can apply to indoor environments as well) while the 24 h CO AQG (7 mg/m$^3$ or 6.2 ppm) is an indoor limit; both are designed to protect people from adverse health effects associated with chronic exposures [11,30]. We also considered the WHO 15 min, 1 h, and 8 h indoor CO AQGs, which are designed to protect residents from CO poisoning resulting from using improperly vented stoves and other faulty appliances indoors [30].

We used box plots, two-sample t-tests, and one-way analysis of variance to explore differences in natural log transformed indoor $PM_{2.5}$ and CO concentrations, and $PM_{2.5}$ I/O ratios, by selected household characteristics shown in the literature to influence indoor air pollutant concentrations. We categorized household characteristics as follows: number of persons (2–4 vs. 5–8), number of rooms (1 vs. 2–4), kitchen volume (m$^3$, continuous measurement), total external doors and windows (1–2 vs. 3–7), rug/carpet floor covering (no vs. yes), combustion activities (i.e., use of cooking fuels [LPG, ethanol, electricity, kerosene], candles, or mosquito repellent; no vs. yes) during air sampling, and the presence of environmental tobacco smoke (i.e., cigarette and/or marijuana smoke in the house; no vs. yes) during air sampling. We also used two-sample t-tests to explore differences in outdoor $PM_{2.5}$ concentrations by participant-observed outdoor air pollution near the home. The criterion employed for statistical significance was $p \leq 0.05$. In sub-analyses, we divided the $PM_{2.5}$ I/O ratios into two groups at the median and repeated the analyses to see whether the household characteristics associated with higher indoor $PM_{2.5}$ differed between homes that were potentially less infiltrated from outdoor $PM_{2.5}$ (i.e., homes with I/O ratios less than the median) compared to homes that were potentially more infiltrated (i.e., homes with I/O ratios greater than the median). Last, we plotted 24 h mean indoor and outdoor T and RH by sample date, and visually explored associations between 24 h mean indoor and outdoor T and RH and indoor and outdoor $PM_{2.5}$ and CO using scatter plots with locally estimated scatterplot smoothing.

## Estimating air change rates for a subset of homes

For a subset of homes with 24 h CO time series with CO decay events, we estimated air change rates (i.e., the number of times a space's air volume is completely removed and replaced per hour), using the decay of indoor CO concentrations produced by combustion activities. This is a common measure of household ventilation [31]. For time series with multiple

decay events, we selected the event associated with afternoon or evening activities (1500–2300 h). Following Batterman (2017) [37], we fit a linear model to the log transformed CO concentrations during the period of CO decay, where the slope of the regression model is the air change rate (h⁻¹). We fit scatter plots and calculated Spearman ρ's between estimated air change rates and indoor $PM_{2.5}$ and CO concentrations. We also compared calculated air change rates to the ASHRAE and EPA recommended rate (0.35 h⁻¹) [38].

### Inclusivity in global research

Additional information regarding the ethical, cultural, and scientific considerations specific to inclusivity in global research is included in the Supporting Information (S1 Checklist).

## Results

### Demographic and household characteristics

At enrollment, mothers who participated in air sampling had a median age of 27 years (interquartile range (IQR), 24–31 years); most (67%) reported having secondary-level education and 33% reported being employed (Table 1). The median monthly household rent was 4,000 Kenya Shillings (IQR: 3,000–5,500) (~33 United States Dollars). The median infant age (n = 47) was 11.8 months (IQR 6.8-16.5 months); one home was assessed 1.7 months before the baby was born. Half of the participants lived in a one-room dwelling (50%), 42% lived in rooms with one window, 85% lived in houses with only one door, and 14 (29%) lived in a house with > 4 residents. Virtually all homes sampled were in the residential units of multi-story buildings, with masonry stone walls and galvanized iron sheet or reinforced concrete slab roofing and consisted of 1–2 rooms with ventilation provided by 1–2 windows and doors.

Air measurements were conducted at 48 homes between July 13, 2022, and June 6, 2024. We summarized descriptives for demographics, cooking behavior and household characteristics of the air sampling participants versus the full ABC cohort (S1 Table). The air sampling participants had a higher proportion with primary education or less (16.7%) vs the full cohort (26.0%). There were also more ethanol users (31.3% vs 17.5%) and fewer kerosene users (37.5% vs 54.0%) compared to the full cohort.

### Self-reported air pollution exposures during air sampling

During the air sampling period, the most commonly used fuel inside the homes was LPG (67%), followed by ethanol (19%) and kerosene (10%) (Table 2). Five participants (10%) reported using electricity and no other fuels. Other reported indoor combustion sources were candles (19%) and mosquito repellent (8%). No participants reported using incense or kerosene lamps or burning rubbish inside the home. Some reported exposure to smoke from their own outdoor cooking (4%) or a neighbor's (15%), while all said they were exposed to other cooking smoke, and 98% said they were exposed to vehicle smoke emissions from sources near home. Other commonly reported outdoor air pollution sources were smoke from the dumpsite (40%) and other rubbish burning (71%), dust from unpaved roads (85%) and construction dust (31%). One participant (2%) reported exposure to smoke from charcoal/brickmaking and none reported exposure to industrial/factory smoke.

### Air measurements - completeness and precision

Among N = 48 homes, 39 (81%) had valid indoor $PM_{2.5}$ and 39 (81%) had valid outdoor $PM_{2.5}$ measurements. Samples were missing or invalid because of security concerns or lack of space (n = 5 outdoor), negative filter mass (n = 1 indoor), RH difference for pre vs. post weigh sessions > 10% (n = 7 indoor, n = 5 outdoor), and sampling duration 33% of target (n = 1 indoor). There were 47 (98%) valid indoor and 41 (85%) valid outdoor CO measurements. Invalid CO measurements were due to battery/equipment failure (n = 1 indoor, n = 2 outdoor). All valid indoor and outdoor $PM_{2.5}$ samples

**Table 1.  Selected demographic and household characteristics of study participants (N=48).**

| Characteristic[a] | Detail | Frequency or median | Percent or IQR |
|---|---|---|---|
| **Socio-demographics at study entry** | | | |
| Infant age, months[b] (n=47) | Median | 11.8 | 6.8-16.5 |
| Maternal age, years | Median | 27 | 24-31 |
| Maternal education level | Primary | 8 | 16.7 |
| | Secondary | 32 | 66.7 |
| | University or college | 8 | 16.7 |
| Maternal employment status | Homemaker or none | 30 | 62.5 |
| | Daily wage | 6 | 12.5 |
| | Small business | 6 | 12.5 |
| | Monthly salary | 4 | 8.3 |
| Monthly household rent (KES) (n=41) | Median | 4,000 | 3,000-5,500 |
| **Household characteristics at air sampling** | | | |
| Type of housing | Small bungalow (single unit) | 1 | 2.1 |
| | Multi-unit dwelling (flats) | 47 | 97.9 |
| Number of persons in household | 2-4 | 34 | 70.8 |
| | 5-8 | 14 | 29.2 |
| Number of rooms | 1 | 24 | 50.0 |
| | 2 | 16 | 33.3 |
| | 3 or more | 8 | 16.7 |
| Persons per room | | 3 | 2-4 |
| Water source | Borehole/rainwater/river | 0 | 0 |
| | Water vendor | 2 | 4.2 |
| | Piped water outside house | 40 | 83.3 |
| | Piped water inside house | 8 | 16.7 |
| Type of toilet | Pit latrine | 0 | 0 |
| | Flush | 48 | 100 |
| Shared toilet | Yes | 42 | 87.5 |
| | No | 6 | 12.5 |
| Roof material | Metal sheets | 33 | 68.8 |
| | Concrete | 15 | 31.3 |
| Wall material | Metal sheets | 1 | 2.1 |
| | Stone | 47 | 97.9 |
| Floor material | Cemented | 34 | 70.8 |
| | Ceramic tiles | 15 | 31.3 |
| **House ventilation features** | | | |
| Number of external windows | 0 | 1 | 2.1 |
| | 1 | 20 | 41.7 |
| | 2 | 19 | 39.6 |
| | 3 or more | 8 | 16.7 |
| Number of external doors (n=47) | 1 | 41 | 85.4 |
| | 2 or more | 6 | 12.5 |
| External openings (windows and doors) per room (n=47) | Median number | 2 | 1.5-2 |
| Volume of cooking room (cubic meters) | Median | 24.5 | 17.9-30.2 |
| **Household behaviors related to air pollution** | | | |
| Rug or carpet floor covering | No | 22 | 45.8 |
| | Yes | 26 | 54.2 |

*(Continued)*

**Table 1.** (Continued)

| Characteristic[a] | Detail | Frequency or median | Percent or IQR |
|---|---|---|---|
| Indoor combustion/burning (cooking fuels) | Wood | 1 | 2.1 |
| | Charcoal | 1 | 2.1 |
| | Kerosene | 18 | 37.5 |
| | Ethanol (Koko fuel) | 15 | 31.3 |
| | Liquefied petroleum gas | 40 | 83.3 |
| | Electricity | 5 | 10.4 |

IQR = interquartile range. Total percentages may exceed 100% as categories were not mutually exclusive. [a]N = 48 unless otherwise stated. [b]At air sampling.

**Table 2.** Combustion related indoor and nearby outdoor activities reported during the 24 hour air sampling period.

| Activity | | Detail | Frequency (N = 48) | Percent |
|---|---|---|---|---|
| **Indoor sources of air pollution** | | | | |
| Cooking fuel used during air sampling[a] | | Wood | 1 | 2.1 |
| | | Charcoal | 1 | 2.1 |
| | | Kerosene | 5 | 10.4 |
| | | Ethanol | 9 | 18.8 |
| | | LPG | 32 | 66.7 |
| | | Electricity | 5 | 10.4 |
| Other activities | | Burning mosquito repellent | 4 | 8.3 |
| | | Burning candles | 9 | 18.8 |
| | | Cigarette use | 2 | 4.2 |
| | | Marijuana use | 4 | 8.3 |
| **Outdoor sources of air pollution** | | | | |
| Outdoor smoke from cooking close to the house | | Own cooking | 2 | 4.2 |
| | | Neighbor's cooking | 7 | 14.6 |
| Other activities contributing to air pollution away from the house but within one kilometer | Outdoor combustion sources | Other cooking | 48 | 100 |
| | | Vehicle smoke | 47 | 97.9 |
| | | Dumpsite | 19 | 39.6 |
| | | Rubbish burning | 34 | 70.8 |
| | | Industry or factory smoke | 1 | 2.1 |
| | | Charcoal or brick making | 1 | 2.1 |
| | Non-combustion sources | Unpaved roads | 41 | 85.4 |
| | | Construction dust | 15 | 31.2 |

IQR = interquartile range. LPG = liquefied petroleum gas. Total percentages may exceed 100% as categories were not mutually exclusive. [a]Note: more than one fuel type may have been used.

represented the full 24 h because > 80% of the 24 h period was captured: indoor range 19.6-24 h, outdoor range 19.6-24 h. The 47 valid indoor CO samples ranged from 24.0-24.2 h and the 41 valid outdoor CO samples ranged from 16.1-24.2 h. Two outdoor CO samples (16.1 h, 16.8 h) shorter than the target of 24 h ± 20% were analyzed as is. The $PM_{2.5}$ relative percent differences were 29% and 114% for the two indoor duplicate pairs and 11% for the outdoor pair. The relative percent difference for the two 24 h mean indoor CO duplicate pairs were 78% and 9%; for the outdoor duplicate pair,

all measurements from both sensors were 0.0 ppm. For the two pairs of indoor CO duplicates, the slopes of the 1 min smoothed data were 1.1 and 1.4, intercepts were 0.0 and 0.1 ppm CO, $R^2$'s were 1.0 and 0.9, and root mean squared errors were 0.2 and 0.4 ppm.

## PM$_{2.5}$ concentrations

The 24 h indoor PM$_{2.5}$ concentrations ranged from 12.8-519.6 ug/m$^3$ with a median of 39.9 ug/m$^3$ (IQR 28.9-63.8 ug/m$^3$). Outdoor concentrations ranged from 2.6-68.2 ug/m$^3$ with a median of 23.2 ug/m$^3$ (IQR 16.7-33.5 ug/m$^3$) (Table 3). Indoor and outdoor PM$_{2.5}$ concentrations were correlated ($\rho = 0.42$, $p = 0.01$). The three participants with complete indoor PM$_{2.5}$ measurements who reported using only electricity and no other fuels during air sampling had lower indoor PM$_{2.5}$ than those who reported burning LPG, ethanol, kerosene, and/or wood indoors (geometric mean (geometric SD) 21.8 ug/m$^3$ (1.3) vs. 47.9 ug/m$^3$ (2.1), respectively (S2 Table). Indoor PM$_{2.5}$ did not differ significantly by number of persons, number of rooms, kitchen volume, total external windows and doors, rug/carpet floor covering, or kerosene, LPG, or ethanol fuel combustion during air sampling. Likewise, indoor PM$_{2.5}$ did not differ by environmental tobacco smoke in the home, or burning mosquito repellent, or candles during air sampling. Outdoor PM$_{2.5}$ did not differ significantly by reported exposure to smoke from their own outdoor cooking or a neighbors' cooking, or smoke from the dumpsite or rubbish burning, or dust from unpaved roads or construction (S3 Table).

For most homes (77%), the indoor PM$_{2.5}$ concentration was higher than the outdoor concentration (Fig 1). PM$_{2.5}$ concentrations exceeded the WHO 24 h AQG of 15 ug/m$^3$ in 79% of outdoor and 97% of indoor measurements. Two homes reporting no indoor combustion of any kind during air sampling had indoor PM$_{2.5}$ concentrations at or above the WHO 24 h AQG (15.5 and 25.8 ug/m$^3$, respectively) [11].

## PM$_{2.5}$ I/O ratios

For the 36 homes with complete indoor and outdoor PM$_{2.5}$ measurements, PM$_{2.5}$ I/O ratios ranged from 0.5-18.3, with a median of 1.6 (IQR 1.1-3.0). When we categorized homes into those with PM$_{2.5}$ I/O ratios ≤ or> than the median ratio, indoor PM$_{2.5}$ did not differ significantly by reported exposure to smoke from any source considered, for either category, with two exceptions (S4 and S5 Tables). Among the homes with I/O ratios ≤ 1.6, indoor PM$_{2.5}$ differed by reported exposure to smoke from rubbish burning, with geometric mean indoor PM$_{2.5}$ 36.3 (1.6) ug/m$^3$ in the exposed (n = 15) vs. 19.3 (1.3) ug/m$^3$ in the unexposed group (n = 3). Among the homes with I/O ratios >1.6, indoor PM$_{2.5}$ differed by reported exposure to construction dust, with geometric mean indoor PM$_{2.5}$ 35.1 (6.3) ug/m$^3$ in the exposed (n = 5) vs. 79.6 (2.5) ug/m$^3$ in the unexposed group (n = 13).

## CO concentrations

The 24 h mean indoor CO concentrations were higher than the outdoor concentrations at all homes (Fig 1). Indoor concentrations (n = 47) ranged from 0.0-33.9 ppm (median 0.7, IQR 0.3-2.4 ppm) and outdoor concentrations (n = 41) ranged

**Table 3. 24 h indoor and outdoor PM$_{2.5}$ (ug/m$^3$) and CO (ppm) concentrations in the infants' homes (n = 48).**

| Pollutant | Units | N | Geo. mean (GSD)[a] | Range | Median | P25 | P75 | P95 |
|---|---|---|---|---|---|---|---|---|
| Indoor PM$_{2.5}$ | ug/m$^3$ | 39 | 45.1 ± 2.1 | 12.8-519.6 | 39.9 | 28.9 | 63.8 | 219.7 |
| Outdoor PM$_{2.5}$ | ug/m$^3$ | 39 | 22.7 ± 1.82 | 2.6-68.2 | 23.2 | 16.7 | 33.5 | 49.5 |
| Indoor CO | ppm | 47 | 2.9 ± 5.9 | 0.0-33.9 | 0.7 | 0.3 | 2.4 | 7.7 |
| Outdoor CO | ppm | 41 | 0.1 ± 0.2[a] | 0.0-1.0 | 0.0 | 0.0 | 0.1 | 0.5 |

N = number of homes. Geo. = geometric. GSD = geometric standard deviation. P = percentile. PM$_{2.5}$ = particulate matter < 2.5 um. CO = carbon monoxide.
[a]Except where noted; otherwise, the arithmetic mean and standard deviation are provided.

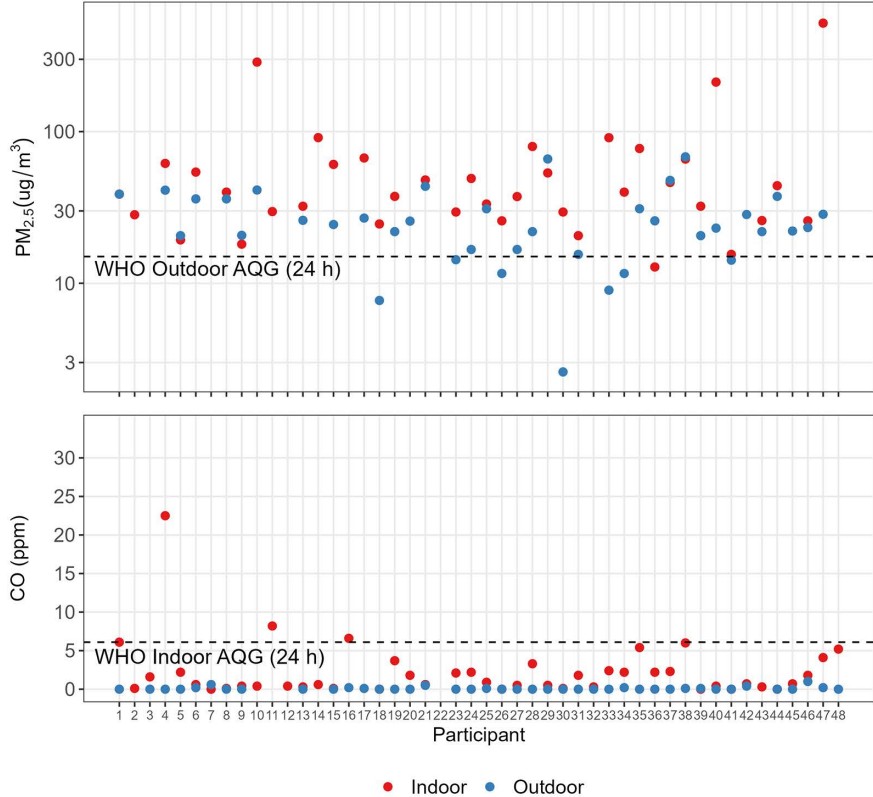

**Fig 1. Measured 24 h PM$_{2.5}$ (ug/m$^3$) (top) and 24 h mean CO (ppm) (bottom) concentrations (Note: top panel y-axis is log base 10 scale for better resolution of indoor vs. outdoor measurements).**

from 0.0-1.0 ppm (median 0.0, IQR 0.0-0.1 ppm) (Table 3). The indoor PM$_{2.5}$ and CO concentrations were correlated ($\rho = 0.37$; $p = 0.02$).

Four of 47 homes (9%) had 24 h mean indoor CO concentrations exceeding the WHO AQG of 6.2 ppm [30]. Three of these reported using ethanol for cooking during air sampling while one used kerosene; none reported using LPG or electricity. The two homes reporting no indoor combustion of any kind during air sampling had 24 h mean indoor CO concentrations of 0.0 ppm and 1.8 ppm. Indoor CO did not differ by number of persons, number of rooms, kitchen volume, total external windows and doors, or use of electricity, LPG, or ethanol during air sampling (S6 Table). The kerosene using homes (n = 5) had higher 24 h mean CO compared to the 42 non-kerosene homes (geometric mean 3.3 (3.1) ppm vs. 0.6 (1.9) ppm, respectively), as did the mosquito repellent homes (n = 4) vs. the non-mosquito repellent homes (n = 43) (geo-metric mean 2.5 (10.7) ppm vs. 0.7 (2.0) ppm, respectively). Indoor CO did not differ significantly by tobacco or marijuana smoke presence or candle use.

For many homes sampled, the 24 h means did not reflect shorter-duration episodes of high indoor CO (i.e., ≥ 50 ppm) seen in the 1 min data. These episodes generally coincided with cooking and mealtimes, that is, morning, after-noon, and evening hours, and CO gradually decreased to as low as 0.0 ppm afterwards. To illustrate, Fig 2 shows the 1 min smoothed CO concentrations in three selected homes. Each panel shows higher indoor compared to outdoor levels, which were generally negligible, and two or three high CO episodes lasting several hours. Two (Fig 2A and 2B) used ethanol during air sampling while the third (Fig 2C) used kerosene. The Fig 2A home also burned mosquito repellent, the Fig 2B home reported presence of environmental tobacco smoke, and the Fig 2C home burned candles

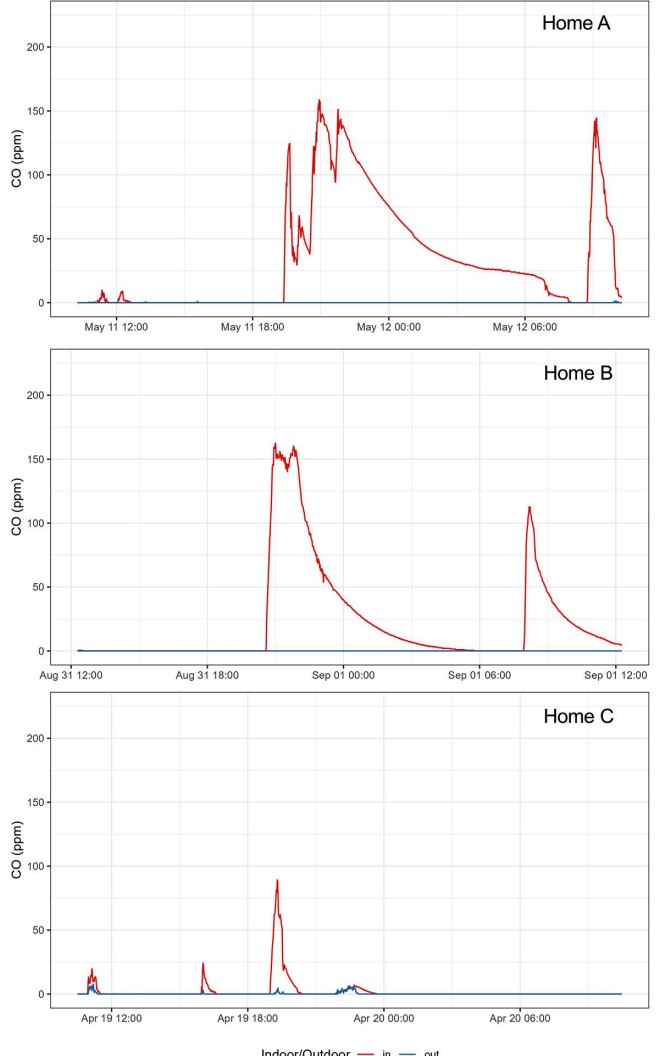

**Fig 2. One minute smoothed indoor and outdoor CO concentrations during sampling at three homes.**

during air sampling. The Fig 2A and 2B homes exceeded the WHO 24 h AQG while the Fig 2C home did not. Additionally, six homes (13%) (including Fig 2A and 2B homes) exceeded the 1 h WHO AQG (30.6 ppm) and three (6%) (including Fig 2A and 2B homes) exceeded the 15 min WHO AQG (87.3 ppm) one or more times during the 24 h period [11,30].

## Associations with temperature and relative humidity

The median indoor (n = 48) and outdoor (n = 44) T were 25.1°C (IQR 23.8-26.5°C) and 23.0°C (IQR 21.8-24.3°C), respectively, while the median indoor (n = 48) and outdoor RH (n = 44) were 65.1% (IQR 57.5-70.3%) and 66.8% (IQR 59.1-69.9%). In the scatter plots with smooth curve fits, associations were not visually apparent between T or RH and indoor or outdoor $PM_{2.5}$ (S1 Fig) or indoor or outdoor CO. (S2 Fig).

## Estimated air change rates

Twenty-three homes had 24 h time-series with one or more CO decay events suitable for air change rate estimation. Estimated air change rates for these homes ranged from 0.3-7.7 with a median of 2.1 (IQR 1.3-3.6). The $R^2$ for these linear decay models ranged from 0.9-1.0 (S7 Table). Among the 23 homes, we did not observe a significant correlation between estimated air change rates and indoor $PM_{2.5}$ ($\rho = 0.15$, $p = 0.6$), but air change rates were inversely associated with 24 h mean CO ($\rho = -0.61$, $p = 0.002$) (S3 Fig). One home had an estimated 0.29 air changes per hour, below the ASHRAE and EPA recommended value of 0.35 $h^{-1}$ [38]. This two-room home also had the highest 24 h mean indoor CO concentration despite having a kitchen volume of 26 $m^3$ (67th percentile) and three external windows and doors.

## Discussion

In this first study of indoor and outdoor $PM_{2.5}$ and CO in infants' homes in a densely populated SSA city [39], we found that indoor $PM_{2.5}$ concentrations exceeded the health-based WHO AQG in 97% of homes even though most reported using clean household fuels (i.e., electricity, LPG, ethanol) during air sampling. Outdoor $PM_{2.5}$ concentrations also exceeded the WHO AQG at 79% of the homes. Other Nairobi studies [4,40–42] also reported high outdoor $PM_{2.5}$ over sampling periods of 8–12 h, indicating urban air quality deterioration potentially due to contributing factors such as population increase, uncontrolled waste burning, unpaved roads, and vehicle exhaust.

In our study, the three homes who reported using electricity and no other fuels during sampling had lower indoor $PM_{2.5}$ compared to homes burning solid or liquid fuels, however their geometric mean still exceeded the WHO AQG, as did indoor $PM_{2.5}$ in the two homes reporting no indoor combustion of any kind during sampling. Bivariate exploratory analyses did not reveal differences in indoor or outdoor $PM_{2.5}$ by any other factor considered, possibly because of small sample sizes.

Indoor and outdoor $PM_{2.5}$ concentrations were moderately correlated, similar to previous studies in other urban residences in India, China, the USA, and Europe [19,26,43,44], possibly indicating the influence of outdoor sources on indoor $PM_{2.5}$. However, the wide range of $PM_{2.5}$ I/O ratios in the 36 homes with complete measurements made it difficult to discern the influence of outdoor $PM_{2.5}$ on indoor levels. When we divided homes into those potentially less infiltrated by outdoor $PM_{2.5}$ vs. those potentially more infiltrated, we found that indoor $PM_{2.5}$ was significantly higher in the potentially less infiltrated homes that reported exposure to smoke from nearby rubbish burning, possibly reflecting the influence of this outdoor source on indoor $PM_{2.5}$ although sample sizes were small (only three participants reported no exposure). Among the potentially more infiltrated homes, indoor $PM_{2.5}$ differed significantly by reported exposure to construction dust, but in the opposite direction from expected (i.e., non-exposed homes had higher indoor $PM_{2.5}$ than exposed homes). It is also theoretically possible that high indoor $PM_{2.5}$ in some homes influenced outdoor levels since we sampled so close (< 1 m) to the outer wall of the home [24]; however, since our participants were using mostly fuels considered to be clean, and not biomass, we did not evaluate this explicitly.

Our results indicate that indoor air quality is a concern for infant health in Nairobi despite the use of clean household fuels. Just over half (53%) of the urban households in the 2022 Kenya Demographic and Health Survey reported having access to clean cooking fuels [39]. Increased access to clean fuels is credited with considerable reductions in mortality and acute respiratory infections in children under 5 years old in recent decades including in SSA [45]. Despite the reported use of clean fuels, the indoor $PM_{2.5}$ concentrations we observed still exceeded the WHO AQG and were consistently higher than outdoor concentrations, likely reflecting a combination of ventilation, infiltration of outdoor $PM_{2.5}$, and indoor combustion activities [2,8,27]. Mutahi et al. (2021) also found higher indoor vs. outdoor 12 h gravimetric $PM_{2.5}$ concentrations in their small study of 15 one room homes in the N'gando informal settlement area of Nairobi in 2019 [4]. Most of the homes we sampled were only one or two rooms, where the infants likely spend much of their time which may lead to high exposures and health burden. The air change rates we estimated for 23 homes generally met the ASHRAE/EPA recommendation, except for one 2 room home (Fig 2B). This was an ethanol using home that also had the highest indoor

CO concentration, suggesting a need for counseling on the importance of opening windows and doors while cooking even when using clean fuels.

Indoor CO levels were generally low over the 24 hours although shorter episodes with concentrations exceeding the WHO 1 h and 15 min AQGs were observed in some homes during typical cooking hours. Outdoor CO concentrations were generally not detectable. The indoor CO concentrations we observed are similar to those reported by Orina et al. (2024) who sampled 71 homes in Mukuru informal settlement, Nairobi, using Lascar CO sensors logging at 1 min intervals [46]. Their 24 h indoor median, 2.9 ppm (IQR 1.2-5.4 ppm), was below the WHO AQG [46]. The indoor CO spikes we observed in some homes are a concern particularly in homes with infants and pregnant mothers. Once inhaled, CO binds to hemoglobin to form carboxyhemoglobin which disrupts tissue oxygenation, leading to poisoning signs and symptoms at carboxyhemoglobin levels > 10% [47]. During gestation, CO crosses the placenta and binds to fetal hemoglobin which has a higher affinity for both oxygen and CO than adult hemoglobin [48]. Fetal hemoglobin is still found in infant circulation during the first year of life; small amounts are present in maternal blood in pregnancy [49]. Although compensatory mechanisms exist to protect fetal, infant and adult brains from hypoxia during acute exposures [30], CO unbound from carboxyhemoglobin after acute or chronic, lower-level exposures can diffuse into nearby tissues causing oxidative stress, inflammation, apoptosis, and other types of damage, including neuronal damage [47,50]. The WHO 15 min, 1 h, and 8 h AQGs are designed to protect residents from CO poisoning resulting from using improperly vented stoves and other faulty appliances indoors, while the 24 h AQG is designed to be protective against health effects in adults from chronic exposures [30]. Safe exposure levels for infants are not currently known. Infants have nearly twice the oxygen consumption rate per unit body weight as adults [51], which likely means they have greater CO absorption per ppm of CO inhaled, thus guidelines aimed at protecting adults may not be protective for infants. Because of the potential for both acute CO poisoning as well as adverse neurodevelopmental and other effects at lower exposure levels [50], it is critical to remind expectant mothers and caregivers of infants to open windows and doors during cooking, even when clean fuels (e.g., ethanol) are used.

## Strengths and limitations

This study provides novel insights into indoor and outdoor $PM_{2.5}$ and CO levels in urban SSA infants' homes. We used gold standard methods to measure indoor and outdoor $PM_{2.5}$ levels and compared both indoor and outdoor $PM_{2.5}$ levels to the WHO outdoor AQG since no indoor AQG currently exists for $PM_{2.5}$ [11]. The Lascar sensors we used to measure CO are commonly used and well-described in household air pollution studies. Field studies have reported that the devices offer a relatively low failure rate and moderate to high correlation when compared to higher quality instruments and other Lascar CO sensors [52], although others noted high relative standard deviations (e.g., 24%) [53]. While CO measurements can degrade over the course of years (e.g., 2–4 years), lower CO concentrations can prolong their useful lifespan. Due to the difficulty of obtaining calibration gases in Nairobi, we relied on span gas checks in the USA to identify potentially malfunctioning sensors and otherwise estimated measurement error according to manufacturer specifications. Future air pollution studies in Nairobi and other SSA locations would benefit from having the capacity to calibrate CO and other gas sensors locally.

Other challenges included the conspicuous nature of the indoor and outdoor air monitors, and other logistical constraints involved in acceptability of home visits, which may have impacted generalizability to the overall cohort and to the Dandora community. For example, security concerns limited outdoor sampling at some of the homes. Mothers in the sampled homes were also slightly more educated, and used a slightly different fuel mix than the full cohort (fewer kerosene users, and more ethanol users), limiting generalizability to the full cohort. Importantly, 83% of households in our air sampling cohort reported use of LPG on some or most days, whereas 53% of urban Kenyan households are estimated to have access to any clean fuel and technologies [9]. Although these observations point towards limited generalizability, they also suggest that urban Kenyans may have even higher exposures to unhealthy air indoors than in our subsample.

Although we followed protocols from prior studies [41,42] on where to set up equipment (i.e., in the main kitchen area, at breathing height), fixed monitors like these do not necessarily capture participants' personal exposures as personal monitors would. Further, we did not measure continuous $PM_{2.5}$ and were not able to capture peak exposure periods as we did with the continuous CO data. Finally, RH changes in the gravimetric laboratory invalidated some $PM_{2.5}$ measurements [33], limiting the sample size. The small sample overall limited our ability to explore factors influencing indoor and outdoor pollutant levels.

## Conclusions

Our study provides novel data on $PM_{2.5}$ and CO levels in infant homes in Nairobi that could help inform the development of air quality regulations for $PM_{2.5}$ and CO in urban residential areas of Kenya. Both the indoor and outdoor $PM_{2.5}$ concentrations we measured were often above WHO AQGs, including in homes using clean fuels. These findings highlight the importance of improving indoor air quality, given that caregivers and infants spend much of their time indoors, which increases their exposures and potential health risks. The limited data on air quality in SSA calls for expanded studies on exposures to air pollution among children residing in urban settings in order to develop effective interventions, build public awareness, and inform policy.

## Supporting information

**S1 Table. Select demographic and household characteristics at enrollment of air sampling participants and the full ABC cohort.**
(DOCX)

**S2 Table. Tests of differences in indoor $PM_{2.5}$ concentrations by selected household characteristics and combustion activities during air sampling in a subsample of 39 homes.**
(DOCX)

**S3 Table. Tests of differences in outdoor $PM_{2.5}$ concentrations by participant reported exposure to air pollution from selected outdoor sources during air sampling in a subsample of 39 homes.**
(DOCX)

**S4 Table. Tests of differences in indoor $PM_{2.5}$ concentrations by selected household characteristics and combustion activities during air sampling in a subsample of 36 homes with complete indoor and outdoor $PM_{2.5}$ data, by $PM_{2.5}$ I/O ratio group.**
(DOCX)

**S5 Table. Tests of differences in indoor $PM_{2.5}$ concentrations by participant reported exposure to air pollution from selected outdoor sources during air sampling in a subsample of 36 homes with complete indoor and outdoor $PM_{2.5}$ data, by $PM_{2.5}$ I/O ratio group.**
(DOCX)

**S6 Table. Tests of differences in 24 h mean indoor CO concentrations by selected household characteristics and combustion activities during air sampling in a subsample of 47 homes.**
(DOCX)

**S7 Table. Estimated air changes per hour with the corresponding measured 24 h $PM_{2.5}$ ($ug/m^3$) and CO (ppm) concentrations in a subsample of 23 homes.**
(DOCX)

**S1 Fig. 24 h mean temperature (°C) and relative humidity (%) vs. 24 h PM$_{2.5}$ (ug/m$^3$) in a subsample of 39 homes.**
(DOCX)

**S2 Fig. 24 h mean temperature (°C) and relative humidity (%) vs. 24 h mean CO (ppm) in a subsample of homes (n = 47 indoor, n = 41 outdoor).**
(DOCX)

**S3 Fig. Scatter plots and Spearman correlations between estimated air changes per hour and natural log indoor PM$_{2.5}$ (top panel) and 24 h mean CO (bottom panel) concentrations in a subsample of 23 homes.**
(DOCX)

**S1 Checklist. Inclusivity in global research.**
(DOCX)

**S1 Data. Anonymized participant data and data dictionaries.**
(XLSX)

## Acknowledgments

We thank the participating mothers and their families, and the following team members for their meaningful contributions: Judith Adhiambo, Brendah Isavwah, James Lele, Margaret Murathi, Charity Muthui, Laura Mwangi, Perpetual Nyaguthi, Electine Oyuga, and Lewis Olweywe. We also thank Erica Wetzler and Tim Gould of UW for their useful technical support. Finally, we thank the Atmospheric Deposition Networks Laboratory in the Department of Chemistry and the Department of Pediatrics and Child Health at the University of Nairobi, Kenyatta National Hospital, and the Dandora II Health Centre for operational and institutional support.

## Author contributions

**Conceptualization:** Christopher Zuidema, Edmund Y. W. Seto, Catherine J. Karr, Elizabeth Maleche-Obimbo, Sarah Benki-Nugent.

**Data curation:** Emily Adhiambo, Anne M. Riederer.

**Formal analysis:** Christopher Zuidema, Anne M. Riederer.

**Funding acquisition:** Edmund Y. W. Seto, Catherine J. Karr, Elizabeth Maleche-Obimbo, Sarah Benki-Nugent.

**Investigation:** Vincent K. Kipter, Bernard Makau, Pricilla Wanini Edemba.

**Methodology:** Vincent K. Kipter, Faridah H. Were, Michael J. Gatari, Christopher Zuidema, Edmund Y. W. Seto, Orly D. Stampfer, Pricilla Wanini Edemba, Julian D. Marshall, Timothy V. Larson, Sarah Benki-Nugent, Anne M. Riederer.

**Project administration:** Faridah H. Were, Orly D. Stampfer, Pricilla Wanini Edemba, Catherine J. Karr, Elizabeth Maleche-Obimbo, Sarah Benki-Nugent.

**Resources:** Orly D. Stampfer, Pricilla Wanini Edemba, Anne M. Riederer.

**Software:** Christopher Zuidema, Emily Adhiambo, Anne M. Riederer.

**Supervision:** Faridah H. Were, Michael J. Gatari, Edmund Y. W. Seto, Orly D. Stampfer, Barbra A. Richardson, Anne M. Riederer.

**Validation:** Christopher Zuidema, Anne M. Riederer.

**Visualization:** Christopher Zuidema, Anne M. Riederer.

**Writing – original draft:** Vincent K. Kipter, Christopher Zuidema, Sarah Benki-Nugent, Anne M. Riederer.

**Writing – review & editing:** Faridah H. Were, Michael J. Gatari, Christopher Zuidema, Edmund Y. W. Seto, Orly D. Stampfer, Barbra A. Richardson, Julian D. Marshall, Timothy V. Larson, Catherine J. Karr, Elizabeth Maleche-Obimbo, Sarah Benki-Nugent, Anne M. Riederer.

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
