## [Decision Letter · Decision Letter 0]

30 Jun 2025

PGPH-D-25-00648

Indoor and outdoor fine particulate matter and carbon monoxide concentrations in homes of infants in Nairobi, Kenya

Dear Dr. Riederer,

Thank you for submitting your manuscript to PLOS Global Public Health. After careful consideration, we feel that it has merit but does not fully meet PLOS Global Public Health’s publication criteria as it currently stands. Therefore, we invite you to submit a revised version of the manuscript that addresses the points raised during the review process.

• A rebuttal letter that responds to each point raised by the editor and reviewer(s). You should upload this letter as a separate file labeled 'Response to Reviewers'.

We look forward to receiving your revised manuscript.

Kind regards,

Changwoo Han, M.D., Ph.D.

Academic Editor

Journal Requirements:

2. We do not publish any copyright or trademark symbols that usually accompany proprietary names, eg (R), (C), or TM (e.g. next to drug or reagent names). Please remove all instances of trademark/copyright symbols throughout the text, including ® on page 7 and 8.

3. In the online submission form, you indicated that [The data that supports the findings of this study are available from the corresponding author upon reasonable request.].

a. In a public repository,

b. Within the manuscript itself, or

c. Uploaded as supplementary information.

Additional Editor Comments (if provided):

Please revise the manuscript based on the comments provided by Reviewers

Reviewers' comments:

Reviewer's Responses to Questions

**Comments to the Author**

1. Does this manuscript meet PLOS Global Public Health’s publication criteria? Is the manuscript technically sound, and do the data support the conclusions? The manuscript must describe methodologically and ethically rigorous research with conclusions that are appropriately drawn based on the data presented.? Is the manuscript technically sound, and do the data support the conclusions? The manuscript must describe methodologically and ethically rigorous research with conclusions that are appropriately drawn based on the data presented.

Reviewer #1: Yes

Reviewer #2: Yes

2. Has the statistical analysis been performed appropriately and rigorously?

Reviewer #1: Yes

Reviewer #2: No

3. Have the authors made all data underlying the findings in their manuscript fully available (please refer to the Data Availability Statement at the start of the manuscript PDF file)?

The PLOS Data policy requires authors to make all data underlying the findings described in their manuscript fully available without restriction, with rare exception. The data should be provided as part of the manuscript or its supporting information, or deposited to a public repository. For example, in addition to summary statistics, the data points behind means, medians and variance measures should be available. If there are restrictions on publicly sharing data—e.g. participant privacy or use of data from a third party—those must be specified.requires authors to make all data underlying the findings described in their manuscript fully available without restriction, with rare exception. The data should be provided as part of the manuscript or its supporting information, or deposited to a public repository. For example, in addition to summary statistics, the data points behind means, medians and variance measures should be available. If there are restrictions on publicly sharing data—e.g. participant privacy or use of data from a third party—those must be specified.

Reviewer #1: Yes

Reviewer #2: Yes

4. Is the manuscript presented in an intelligible fashion and written in standard English?

Reviewer #1: Yes

Reviewer #2: Yes

Reviewer #1: I have gone through the MS entitled "Indoor and outdoor fine particulate matter and carbon monoxide concentrations in homes of infants in Nairobi, Kenya" thoroughly and found it very interesting. Before publication few changes needed.

1. Title of the MS is not representable for the whole study. Please modify as per the main objectives.

2. Add all environmental sample size in methodology section.

3. Also include the how the study came up with the 400 sample size.

4. Conclusion part I feel less informative, please improve.

5. Objectives of the study should be well explain in the MS.

Reviewer #2: Thank you for giving me the opportunity to review this paper. I will give you some opinions and suggestions and hope the manuscript will be improved.

[Abstract]

Although the abstract mentions an assessment of air pollution and neurodevelopmental outcomes in children, the results and conclusions do not address neurodevelopmental outcomes. Since the manuscript does not cover this topic in detail,

I recommend removing references to neurodevelopmental outcomes to avoid confusion.

The authors evaluated exceedances of indoor and outdoor PM2.5 and CO concentrations based on WHO air quality guideline (AQG) levels.

However, it is important to note that WHO AQGs primarily focus on outdoor air quality, and more stringent standards may be needed for indoor environments.

The authors should consider discussing this limitation in the Discussion section and/or reference appropriate indoor air quality benchmarks.

[Introduction]

lines 62-65,

This sentence is somewhat unclear. It may be helpful to revise it for clarity and conciseness.

Both urban and rural children are likely to be exposed to indoor and ambient air pollution, so the contrast being drawn here seems oversimplified.

lines 77-79.

Please specify whether these PM2.5 concentrations are 24-hour, monthly, or annual averages, as this is critical for interpreting the pollution levels.

I recommend that the authors include the geometric mean (GeoMean) and geometric standard deviation (GeoSD) in Table 3 to better characterize the distribution of the exposure data, which are likely to be right-skewed. Additionally, regarding the PM2.5 measurement for ID 47 (519.57 µg/m³), do the authors consider this to be a valid measurement?

It would be helpful to clarify whether this value was verified or assessed for potential measurement error, as it appears to be an extreme outlier.

It would be helpful to include a Spearman correlation matrix of indoor/outdoor PM2.5, CO, temperature, and humidity to better illustrate the relationships between air quality and meteorological variables.

Given the small sample size, conducting statistical tests may be challenging.

I recommend that the authors use non-parametric tests instead of t-tests, as these do not assume normality. Additionally, it would be beneficial to discuss this limitation in the Discussion section to acknowledge its potential impact on the results.

[Results]

The authors should provide a more detailed explanation of how the 48 participants were selected from the overall cohort.

It would be helpful to indicate, in the Methods section, how many participants were excluded from the original cohort of approximately 400, and the reasons for their exclusion.

This information is important for assessing potential selection bias and the generalizability of the findings.

If possible, I suggest adjusting the size and position of the WHO AQG label in Figure 1, as it overlaps with some data points, which may reduce the clarity of the figure.

The authors should discuss the representativeness of the study population included in the analysis.

In addition, potential biases related to the placement of indoor and outdoor air pollution monitors (for PM2.5 and CO) should be considered.

For PM2.5, the lack of continuous measurements limits the ability to capture peak exposure periods.

While CO was measured continuously, a notable limitation is the relatively low proportion of valid measurement time within the 24-hour period.

[Conclusion]

lines 510-511

"Four homes had higher indoor CO concentrations than the AQGs."

As a concluding statement, this may lack clarity and impact.

Providing more context (e.g., total sample size, degree of exceedance, or percentage) would strengthen its significance

**Do you want your identity to be public for this peer review?** For information about this choice, including consent withdrawal, please see our Privacy Policy..

Reviewer #1: No

Reviewer #2: No

---

## [Decision Letter · Decision Letter 1]

23 Jan 2026

PGPH-D-25-00648R1

Indoor and outdoor fine particulate matter and carbon monoxide concentrations in homes of infants in Nairobi, Kenya

Dear Dr. Riederer,

Thank you for submitting your manuscript to PLOS Global Public Health. After careful consideration, we feel that it has merit but does not fully meet PLOS Global Public Health’s publication criteria as it currently stands. Therefore, we invite you to submit a revised version of the manuscript that addresses the points raised during the review process.

• A letter that responds to each point raised by the editor and reviewer(s). You should upload this letter as a separate file labeled 'Response to Reviewers'.

We look forward to receiving your revised manuscript.

Kind regards,

Changwoo Han, M.D., Ph.D.

Academic Editor

Journal Requirements:

Additional Editor Comments (if provided):

Reviewers' comments:

Reviewer's Responses to Questions

**Comments to the Author**

Reviewer #2: All comments have been addressed

publication criteria? Is the manuscript technically sound, and do the data support the conclusions? The manuscript must describe methodologically and ethically rigorous research with conclusions that are appropriately drawn based on the data presented.? Is the manuscript technically sound, and do the data support the conclusions? The manuscript must describe methodologically and ethically rigorous research with conclusions that are appropriately drawn based on the data presented.

Reviewer #2: Yes

3. Has the statistical analysis been performed appropriately and rigorously?

Reviewer #2: Yes

4. Have the authors made all data underlying the findings in their manuscript fully available (please refer to the Data Availability Statement at the start of the manuscript PDF file)?

The PLOS Data policy requires authors to make all data underlying the findings described in their manuscript fully available without restriction, with rare exception. The data should be provided as part of the manuscript or its supporting information, or deposited to a public repository. For example, in addition to summary statistics, the data points behind means, medians and variance measures should be available. If there are restrictions on publicly sharing data—e.g. participant privacy or use of data from a third party—those must be specified.requires authors to make all data underlying the findings described in their manuscript fully available without restriction, with rare exception. The data should be provided as part of the manuscript or its supporting information, or deposited to a public repository. For example, in addition to summary statistics, the data points behind means, medians and variance measures should be available. If there are restrictions on publicly sharing data—e.g. participant privacy or use of data from a third party—those must be specified.

Reviewer #2: Yes

5. Is the manuscript presented in an intelligible fashion and written in standard English?

Reviewer #2: No

Reviewer #2: The authors have addressed the concerns I previously raised, but I believe some additional revisions are necessary.

(1) In Table 1-2, some items appear to include overlapping responses, for example, Floor material and Fuels used by the household indoors, and I recommend that the authors add notes or footnotes to clarify these overlaps explicitly.

(2) Lines 496-497 include the sentence "it is plausible that an indoor ~ outdoor AQG", which is unclear and should be rewritten for clarity.

(3) In the Conclusion, while I acknowledge the need for further research, the sentence regarding "Four homes ~ similar contexts" should either be deleted or revised, as it does not effectively highlight the study’s conclusions.

(4) When it comes to the global context, generally, indoor and outdoor PM and CO levels are higher outdoors, but in regions such as Nairobi or Sub-Saharan Africa, indoor concentrations remain high, which may be attributed to factors such as fuel use, socioeconomic conditions, and cultural practices. I recommend emphasizing in the discussion the importance of improving indoor air quality in relation to these findings and noting that since most activities are conducted indoors rather than outdoors, the potential for greater health burdens may arise.

(5) I suggest that the authors consider additional English editing if possible.

**Do you want your identity to be public for this peer review?** For information about this choice, including consent withdrawal, please see our Privacy Policy..

Reviewer #2: No

---

## [Decision Letter · Decision Letter 2]

15 Mar 2026

Indoor and outdoor fine particulate matter and carbon monoxide concentrations in homes of infants in Nairobi, Kenya

PGPH-D-25-00648R2

Dear Riederer,

We are pleased to inform you that your manuscript 'Indoor and outdoor fine particulate matter and carbon monoxide concentrations in homes of infants in Nairobi, Kenya' has been provisionally accepted for publication in PLOS Global Public Health.

Best regards,

Changwoo Han, M.D., Ph.D.

Academic Editor

Reviewer Comments (if any, and for reference):

Reviewer's Responses to Questions

**Comments to the Author**

Reviewer #2: All comments have been addressed

publication criteria? Is the manuscript technically sound, and do the data support the conclusions? The manuscript must describe methodologically and ethically rigorous research with conclusions that are appropriately drawn based on the data presented.? Is the manuscript technically sound, and do the data support the conclusions? The manuscript must describe methodologically and ethically rigorous research with conclusions that are appropriately drawn based on the data presented.

Reviewer #2: Yes

3. Has the statistical analysis been performed appropriately and rigorously?

Reviewer #2: Yes

4. Have the authors made all data underlying the findings in their manuscript fully available (please refer to the Data Availability Statement at the start of the manuscript PDF file)?

The PLOS Data policy requires authors to make all data underlying the findings described in their manuscript fully available without restriction, with rare exception. The data should be provided as part of the manuscript or its supporting information, or deposited to a public repository. For example, in addition to summary statistics, the data points behind means, medians and variance measures should be available. If there are restrictions on publicly sharing data—e.g. participant privacy or use of data from a third party—those must be specified.requires authors to make all data underlying the findings described in their manuscript fully available without restriction, with rare exception. The data should be provided as part of the manuscript or its supporting information, or deposited to a public repository. For example, in addition to summary statistics, the data points behind means, medians and variance measures should be available. If there are restrictions on publicly sharing data—e.g. participant privacy or use of data from a third party—those must be specified.

Reviewer #2: Yes

5. Is the manuscript presented in an intelligible fashion and written in standard English?

Reviewer #2: Yes

Reviewer #2: I thank the authors for their efforts in addressing the reviewers’ comments. I have no further comments.

**Do you want your identity to be public for this peer review?** For information about this choice, including consent withdrawal, please see our Privacy Policy..

Reviewer #2: No
